# Two distinct and separable processes underlie individual differences in algorithm adherence: Differences in predictions and differences in trust thresholds

Achiel Fenneman[1,2]*, Joern Sickmann[2], Thomas Pitz[2], Alan G. Sanfey[3]

**1** Institute for Management Research, Radboud University, Nijmegen, Netherlands, **2** Faculty Society and Economics, Rhein-Waal University of Applied Sciences, Kleve, Germany, **3** Donders Institute for Brain, Cognition and Behaviour, Radboud University, Nijmegen, Netherlands

* a.fenneman@fm.ru.nl

**Data Availability Statement:** The experimental software, raw data and analysis scripts are

## Abstract

Algorithms play an increasingly ubiquitous and vitally important role in modern society. However, recent findings suggest substantial individual variability in the degree to which people make use of such algorithmic systems, with some users preferring the advice of algorithms whereas others selectively avoid algorithmic systems. The mechanisms that give rise to these individual differences are currently poorly understood. Previous studies have suggested two possible effects that may underlie this variability: users may differ in their predictions of the efficacy of algorithmic systems, and/or in the relative thresholds they hold to place trust in these systems. Based on a novel judgment task with a large number of within-subject repetitions, here we report evidence that both mechanisms exert an effect on experimental participant's degree of algorithm adherence, but, importantly, that these two mechanisms are independent from each-other. Furthermore, participants are more likely to place their trust in an algorithmically managed fund if their first exposure to the task was with an algorithmic manager. These findings open the door for future research into the mechanisms driving individual differences in algorithm adherence, and allow for novel interventions to increase adherence to algorithms.

## Introduction

The emergence of the modern internet, of smartphone technology, and of an unparalleled availability of data now allow for previously unimaginable levels of decision-support systems. These systems allow for tailored algorithmic advice in a wide range of topics, including (but not limited to) recommendation systems (e.g. Youtube, Netflix and Amazon), navigation (e.g. Google Maps), the suggestion of third-party services (e.g. Uber, Lyft, Airbnb), potential romantic partners (e.g. Tindr), and increasingly also target high-impact decisions such as interpreting medical data. The societal impact of such algorithmic systems is growing with incredible rapidity, and recent developments have moved beyond the mere provision of advice

available in the OSF repository at https://osf.io/ua62h/.

**Funding:** The author(s) received no specific funding for this work.

**Competing interests:** The authors have declared that no competing interests exist.

in favor of full reliance on algorithmic systems, embodied by recent developments in autonomous automotive systems ('self-driving cars') as well as automated financial traders and fund managers ('robo-traders'). Reliance on these systems supplants both user-input and user-control to a significant degree, requiring a high level of trust in the algorithm's efficacy. As the societal impact of such algorithmic support systems increases, so does the importance of understanding when, and why, humans do or do not place their trust in algorithmic support systems.

Despite the dramatic increase in the availability of algorithmic support, there appears to be substantial individual variation across people in their degree of adherence to algorithmic advice and support options (henceforth '*algorithm adherence*'). One the one hand, several recent findings strongly suggest that advice from algorithmic sources is underweighted when compared to advice from fellow humans. This asymmetric discounting of algorithmic advice in particular, termed *algorithm aversion* [1], has been observed in a number of tasks, ranging from forecasting [1–4], visual-detection [5, 6], maze navigation [7], selecting the best joke for an intimate other [8] and medical recommendations [9, 10].

On the other hand, multiple studies [1, 11, 12] have observed that a nontrivial minority of participants actually rely more on algorithmic than on human advice. Experimental participants have been observed to recognize the superior performance of algorithms [1], and more recent studies have discerned no difference in reliance on algorithmic versus human agents in an investment task [13] or trust game [14]. Crucially, recent studies have even observed an inverse effect, with participants relying more on algorithmic advice than human advice dubbed *algorithm appreciation* [15]. Furthermore, recent evidence suggests that participants prefer algorithmically-generated uncertainty over human-generated uncertainty [16]. Taken together, these findings suggest that the level of algorithm adherence may not be uniform across the population but rather fall on a distribution between the two extremes of pure algorithm aversion and pure algorithm appreciation.

On a societal level these individual differences in algorithm adherence have two adverse effects. First, the suboptimal use by a large proportion of users suggests that there is a substantial societal gain left on the table: as algorithmic support systems increase in efficacy, suboptimal use of these algorithms results in users missing out on the benefits. Second, these individual differences in algorithm adherence may result in an increase of societal inequality. As algorithmic support systems offer the potential to increase the user's (professional) productivity, a difference in algorithm adherence may result in a societal and economic rift between algorithmophiles and algorithmophobes. With the ever-increasing impact and efficacy of algorithmic systems, such a rift may result in an unequal distribution of their societal gains. Understanding the mechanisms that give rise to this distribution of algorithm adherence can yield valuable insights in how to remedy this unequal situation. Despite this societal relevance these mechanisms are currently poorly understood. An overview of previous studies on the topic indicates two potential underlying causal paths: differences in algorithm adherence may result from differences in users' predictions of algorithmic efficacy or from differences in user's trust in algorithms.

The first of these two mechanisms suggests that individual differences in algorithm adherence may result from differences in users' *predictions* regarding the efficacy of algorithmic systems [1]. Specifically, users' judgment of efficacy can be influenced by the existence of a perfect automation schema [4–7, 17, 18]. Users who hold such a schema implicitly expect algorithms to provide perfect performance, which can lead to two, opposing, effects when making decisions about such a system. On the one hand users may be less likely to question the performance of algorithms, underweighting any observed errors made by algorithmic agents [19, 20]. This underweighting then results in a comparatively positive prediction of the algorithms'

future performance. On the other hand, for other users any observed errors can break the schema, resulting in higher salience of algorithmic errors and consequentially overweighted impact of these mistakes. This leads to a comparatively *negative* prediction of future efficacy as compared to human performance [5]. In summary, this literature suggests that different subgroups of users are differentially sensitive to the observed errors of algorithmic systems. As no real-world algorithm can deliver 100% perfect performance, the different subgroups will therefore form different, even opposite, predictions of algorithmic efficacy even when both groups observe the exact same objective performance by an (imperfect) algorithm. This is one possible explanation of the observed individual differences in algorithm adherence.

A related but nonetheless distinct alternative body of work suggests that individual variation in algorithm adherence may result from differences in the perceived uncertainty associated with algorithmic systems. Previous research suggests that experimental participants typically describe themselves as having more in common with human rather than algorithmic advisors [4] and report a higher degree of trust in algorithms with more human-like features [21–23]. Such perceived similarity with an advisor is a crucial determinant of adherence to advice [24] and experiencing an algorithm as being dissimilar to oneself has been documented to result in lower reported feelings of both understanding and control over algorithmic processes [8, 15, 25].

These findings are consistent with previously documented differences in users' ability to mentalize the internal states of algorithmic agents: earlier neuroscientific findings suggest that users may employ the same neural substrates to infer the motivations of both human and algorithmic others, although with a significantly reduced intensity [26]. As a result, users may differ in the degree to which they possess a *theory of machine* [15], i.e. their perceived ability to understand and predict the action of algorithms. This line of research suggests that some users may perceive algorithms to be more difficult to understand and hence to be more unpredictable. Such users may not necessarily hold lower predictions for algorithmic than human agents, but instead differ in the minimum predicted outcome required to offset this uncertainty before placing trust. This difference in *trust thresholds* for algorithmic systems (as compared to human systems) provides a second, different, explanation of the observed variance between users' willingness to adhere to algorithmic advice.

In summary, previous research suggests two different mechanisms that may underlie the observed individual differences in algorithm adherence. One the one hand, individual differences in the over- or underweighting of any perceived errors (or general suboptimal performance) by algorithmic systems can result in individual differences in the predicted future performance of algorithmic systems. These differences in *predictions* then lead to the observed individual differences in algorithmic adherence. Alternatively, users may differ in the degree to which they find algorithms to be either familiar and reliable, or unfamiliar and uncertain. Individual differences in this experienced uncertainty results in differences in *trust thresholds*: the minimum predicted performance required before placing trust in algorithmic systems. This variation can result in individual differences in algorithm adherence, even if different users hold otherwise identical predictions of algorithmic and human systems. While both mechanisms have been supported by some limited empirical evidence, it is currently unclear to which degree both mechanisms exist independently from each another. Are these two independent effects, or are they simply two sides of the same coin? The answer to this questions has important implications for both our understanding of human-algorithm interaction, as well practical implications for designing algorithmic systems to maximize user trust and adherence.

Here we report the findings of an online experiment conducted via Amazon Mechanical Turk (AMT). Each participant in this experiment performed four different investment

decision games, in each of which they were given 24 sequential options to invest in a fund managed by either a human (two games) or an algorithm (two games), keeping all fund returns fixed between both manager types. Such an investment decision task was selected as real-world stock returns are inherently stochastic and hence provide an imperfect signal of a fund manager's performance. This imperfect signal forces participants to infer whether their current manager's performance is a function of skill versus luck (for a similar argument see [13]). By varying the nature of the fund manager (human versus algorithmic) but keeping the observed returns constant, we experimentally disentangle each participant's algorithm adherence and differences in predictions and trust thresholds between the two manager types.

The results of the experiment supports the existence of both proposed mechanisms. Individual differences in algorithm adherence are significantly predicted by both the participants' average estimated performance differences of human- versus algorithmically managed funds (i.e. differences in predictions), as well as by the minimal expected performance threshold at which they are willing to invest in a human- versus algorithmically managed fund (i.e. differences in trust thresholds). Crucially, we observe that both of these mechanisms are uncorrelated across the sample; participants who have a high difference in predictions may not have a high trust threshold, and vice versa. This finding provides empirical support for the existence of two independent mechanisms underlying individual differences in algorithm adherence. Furthermore, we observe an order effect such that participants who first encounter an algorithmic fund manager hold lower trust thresholds for algorithmic (as opposed to human-) managed funds, which in turn results in a higher degree of algorithm adherence.

## Materials and methods

### Participants

The online experiment was restricted to participants based in the USA, who had previously completed a minimum of 1000 completed tasks on AMT with a minimal approval rate of 97%. The sample size was based on an a-priori determined fixed budget, with participants being recruited until this budged was depleted. Afterwards no additional participants were recruited. Prior to data collection, the experimental design was approved by the Ethics Committee for Non-Invasive Research on Humans in the Faculty of Society and Economics of the Rhine-Waal University of Applied Sciences. At the start of the experiment (prior to the experimental instructions) participants were informed of the general nature of the experiment and provided written consent to participate in the experiment.

### Experiment design

The experiment was structured around four different *investment games* (Fig 1). At the start of each game the participant received 10 Tokens (the currency used during the experiment) as a starting capital, intended to prevent participants from obtaining negative rewards throughout the game. The participant was then introduced to a new fund manager. Participants were informed that this manager had previously managed a fictive investment fund for 24 discrete periods. Each investment game then consisted of 24 rounds, with each round consisting of three (plus one) phases: a forecasting phase, followed by an investment phase, followed by a feedback phase (occasionally followed by an attention-check phase).

During the first phase of each round participants were instructed to predict that round's fund return by moving a 'forecast' slider ranging from -6% to +6%. The initial position of the slider's thumb was hidden during the first round of each game in order to prevent potentially confounding anchoring effects. In consecutive rounds the initial value of the slider equaled the forecast of the preceding round. The participant was incentivized to be as accurate as possible:

# A experiment procedure

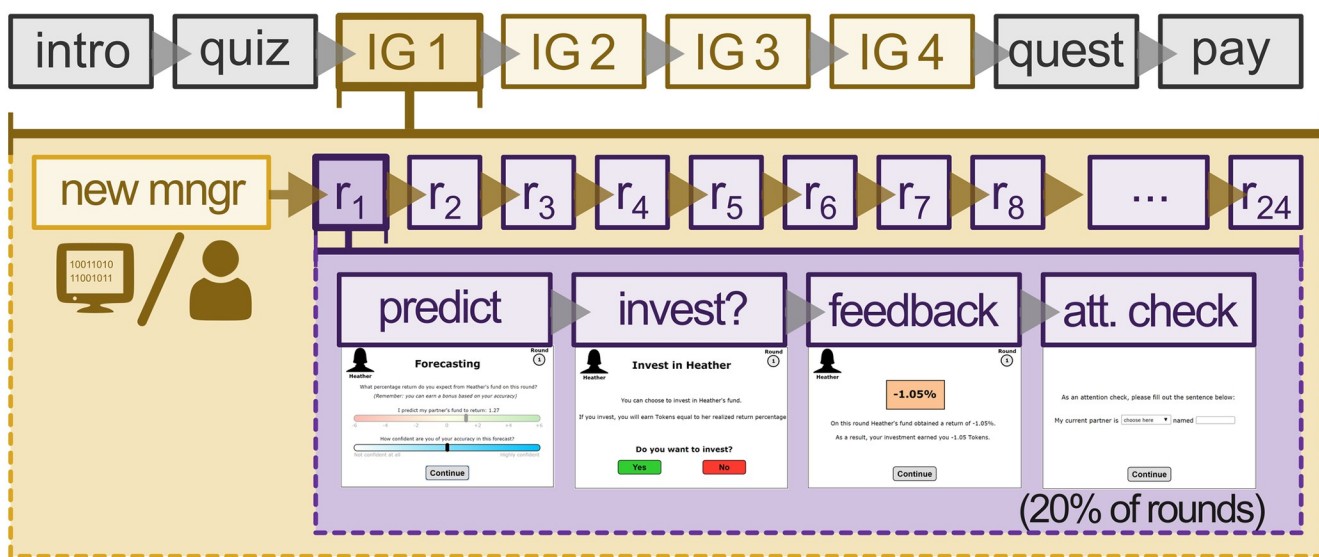

# B observed fund returns

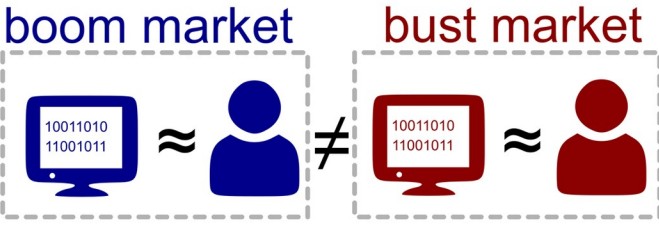

**Fig 1. Schematic overview of the experiment procedure. A**: at the start of the experiment participants were informed of the investment game procedure. These instructions were followed by a mandatory quiz. Upon successful completion of the quiz, the participants continued to the first of four investment games. At the start of each investment game participants were introduced to a new partner (manager). Each game lasted for 24 rounds. Each round consisted of three phases: a prediction, investment and feedback phase. After the feedback phase participants had a 20% probability to encounter an attention check. After the last investment game participants continued to a questionnaire, followed by the payment screen and the conclusion of the experiment **B**: during the four investment games participants were assigned to two human and two algorithmic partners (in both in a boom and bust market).

conditional on the quality of the forecast the participant could earn a bonus of 0.5/1/3 Tokens if the forecast was within 1.0/0.5/0.1 percentage points, respectively. After making their forecast, the participant was asked to rate their (unincentivized) confidence in their forecast via a second slider ranging from "*not confident at all*" to "*highly confident*".

During the second (investment) phase of each round the participant made a binary yes/no decision as to whether to invest in the fund. Finally, during the third (feedback) phase the participant observed the return obtained by the fund. If the participant had decided not to invest on that round, then he/she neither won nor lost any Tokens. If the participant did invest in the fund, they earned Tokens equal to the percentage-point return obtained by the fund. A percentage-point based return (instead of a percentage over an invested amount) was chosen to prevent any confounding income or wealth effects and keeping the returns for each round independent from all other rounds. The feedback screen also provided a summary of the participant's total earnings during the round.

At the end of each round—after the feedback screen but before starting a new round—there was a 20% probability of first transitioning to an attention check phase. These attention checks came in one of two formats. Either the participant was required to recall the percentage return just observed (34% of checks), or to complete the sentence: "my partner is a [human/algorithm] named ___" (66% of checks). This latter attention check served a dual role as manipulation check, ensuring that the participant was aware of the manager type currently engaged with. Participants completed an average of 19.1 attention checks (min = 11, max = 31).

After all 24 rounds of a game had been completed, the participant continued to the next investment game, with a new assigned manager. Feedback on the participant's total earnings was withheld until the end of the experiment. During the instructions, the participant was informed that only one of the four investment games would be selected to determine their performance-based payment for the experiment.

## Procedure

The experiment was conducted online within the AMT environment. At the start of the experiment each participant was provided with a brief introduction to the task, containing the name of the academic institution conducting the experiment, estimates of both expected earnings and the duration of the experiment, a short overview of the different stages of the experiment as well as the exchange rate between Tokens and USD (one Token was worth 0.075 USD). Upon consenting to these terms the participant was provided with detailed instructions for the remainder of the experiment. These instructions were followed by a mandatory quiz, included to ensure that the participant fully comprehended the provided instructions. The participant was only allowed to proceed to the remainder of the experiment if all 16 questions (8 true/false questions, plus 8 numerical input questions) were answered correctly. The participant received feedback on any incorrect answers and was allowed to re-read the instructions and repeat the quiz an unlimited number of times. The quiz included a number of example calculations. In order to prevent anchoring effects, all values used in these calculations were randomized on a participant-level basis. Upon completing the quiz, the participant continued to the first investment game.

In two out of four investment games the fund was managed by a human, while in the other two games the fund was managed by an algorithm. Each human manager was identified by their first name and gender (represented as either a male or female icon). Each algorithm was introduced by its "name": a unique combination of two letters followed by two numbers. This name was included to insure that participants understood that each algorithmic manager represented a unique agent. Throughout the game a visual reminder of the manager's identity was placed in the top-right corner of the screen. This reminder consisted of an icon of a computer (denoting an algorithmic manager) or a human silhouette (denoting a human manager, with a male silhouette denoting male managers and a female silhouette denoting female managers). The manager's name was presented below this icon. In order to mimic real-world scenarios, and in line with previous research, the algorithm's underlying mechanisms were not disclosed to participants.

In order to avoid the use of deception, all observed return sequences where drawn from real human/algorithmic managers. The human data was collected via a previously conducted stimulus pilot experiment. Participants in this pilot experiment (henceforth referred to as "*human fund managers*") completed 24 rounds of a *fund manager game*, during each round of which they could select stock(s) to invest their fund into. Unbeknownst to the human fund managers, all available stocks were selected such that these obtained nearly identical returns.

Human fund managers were not eligible to participate in the main experiment. In order to prevent potentially confounding effects of social preferences, the human fund managers' return rates were determined in a "*cold state*": the investment decisions of participants in the main experiment did not influence the earnings of the human fund managers, nor were the human fund managers informed of these decisions. Finally, in order to prevent framing effects, in the main experiment all fund managers (both human and algorithmic) were referred to as a "partner" throughout the investment games.

Unbeknownst to the participants in the main experiment, the managers of the four investment games were coupled in pairs. Each pair contained one human fund manager (uniquely sampled from a set of 12 male and 12 female managers) and one algorithmic fund manager. The returns of the algorithmically managed fund were constructed such that the sequence of fund's returns were nearly identical (r > 0.97) to that of the matched human-managed fund. The presentation order of managers was pseudo-randomized such that both managers of the same pair would not be presented consecutively. All sequences of returns had a mean return of zero. In order to ensure a sufficient degree of variance in participant's forecasting errors, the returns of one pair of managers (i.e. one human-managed and one algorithm-managed fund) followed a boom market state with high positive returns during the middle rounds of the sequence (see top panels of Fig 2). The fund returns of the other pair of managers (again one human and one algorithm manager) followed a bust market state, with high negative returns during the middle rounds of the sequence (see bottom panels of Fig 2). This procedure ensured that the exact sequence of observed return amounts was (approximately) identical for the human and algorithmic managers, whilst maintaining a degree of unpredictability and variance in the observed returns. Furthermore, in order to control for any unintentional confounds we generated two sequence types. These types differ in the first and last sections of the sequence, such that type II (right panels of Fig 2) was a mirrored version of type I (left panels of Fig 2). In summary: each participant played with four separate managers, one human and one algorithmic manager in a boom sequence, and one human and one algorithm in an inverted bust sequence.

Following the completion of the fourth and final investment game, the participant completed a questionnaire phase containing a number of additional control tasks. These included an incentivized risk-elicitation method and an unincentivized questionnaire. Participants' risk preferences were measured using a hot version of the Bomb Risk Elicitation Task [27]. This task was incentivized such that the participant could earn up to 10 additional Tokens based on the amount of risk the participant was willing to take (as well as an element of luck). This task was followed by a brief questionnaire consisting of items pertaining to the participant's demographics, smartphone usage, their perceived daily-life importance of their smartphone, general opinion on the reliability of computers, the degree to which they would trust a computer to manage specific decision contexts for them and the degree to which participants adhere to algorithmic advice for online videos. Table 1 provides a detailed list of all the items included in the questionnaire.

## Pre-registration and data availability

The statistical procedures to determine a participant's level of algorithm adhesion, differences in predictions, and differences in trust thresholds were preregistered before the final data was analyzed and can hence be construed as being confirmatory in nature. We did not preregister specific hypotheses for the remaining statistical tests. The preregistration report can be accessed online at https://osf.io/sc3jn. All experimental software code, raw data and statistical analysis scripts are available in the OSF repository at https://osf.io/ua62h/.

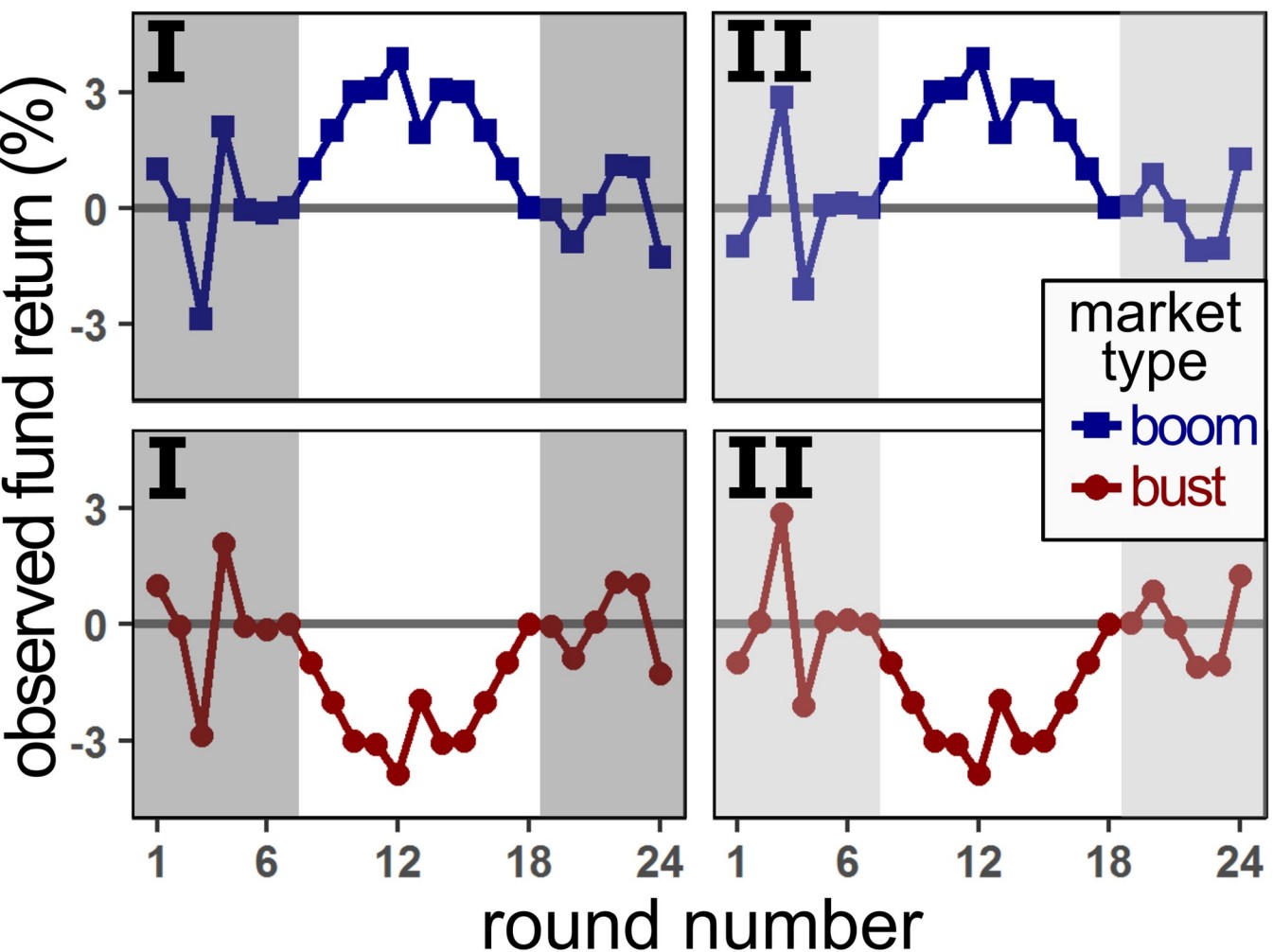

**Fig 2. Observed fund return sequences.** All observed fund return sequences where drawn from 4 different sequences: type (I or II) x market (boom or bust). The two types were mirrored versions of the highlighted regions, whilst the two market types were mirrored versions of the unshaded regions. Each {human, algorithm} pair was assigned a different sequence. Sequences were assigned pseudo-randomly such that each participant always interacted with one human/ algorithm pair in both sequence types and both markets (e.g. boom/I and bust/II, or bust/I and boom/II).

## Results

A total of 119 participants completed the experiment. The median duration of the experiment was approximately 34 minutes, with a mean total compensation of 4.62 USD. Seven participants were excluded from further analysis for failing to provide a correct answer in at least 70% of the attention checks. An additional nine participants were excluded from the analyses based on their observed trust thresholds; eight of these participants had insufficient variance in their investment decisions and one participant displayed an extreme difference in trust thresholds (more than 7 SD difference above the mean of all participants). All analyses reported in the remainder of this study are robust to the inclusion of these eight participants where available. The final sample of 103 included participants was predominantly male (66%, with <1% answering "other") and had a median age of 34 years (SD = 10.15). The majority of participants either completed high-school (41%) or a bachelor's (49%) degree as their highest completed education, with a small minority completing a master's (10%) or doctoral (<1%) degree.

**Table 1. Correlations between questionnaire items and our measure of algorithm adherence.**

| Questionnaire item | n | test | corr. | p |
|---|---|---|---|---|
| **Demographics** | | | | |
| Age | 103 | PR | -0.087 | 0.381 |
| Education | 103 | SR | 0.037 | 0.713 |
| Gender | 103 | OLS | N/A | 0.707 |
| Risk Aversion (BRET) | 103 | PR | -0.151 | 0.129 |
| **Smartphone and App usage** | | | | |
| Smartphone usage | 103 | SR‡ | -0.019 | 0.575 |
| Navigation apps[1] | 98 | SR‡ | 0.14 | *.085†* |
| Freq. switch default route[2] | 89 | SR‡ | 0.006 | 0.477 |
| Ride-sharing apps[1] | 98 | SR‡ | 0.03 | 0.384 |
| Banking and finance apps[1] | 98 | SR‡ | 0.042 | 0.34 |
| Social media apps[1] | 98 | SR‡ | 0.014 | 0.445 |
| Dating apps[1] | 98 | SR‡ | 0.03 | 0.383 |
| Short term housing apps[1] | 98 | SR‡ | 0.116 | 0.127 |
| Personal fitness apps[1] | 98 | SR‡ | -0.012 | 0.546 |
| **Online video suggestions** | | | | |
| Suggestions typically followed | 103 | PR‡ | 0.094 | 0.173 |
| Suggestions typically useful[3] | 103 | PR‡ | -0.004 | 0.518 |
| **Attitudes Towards Computers** | | | | |
| Importance of smartphone | 103 | PR‡ | 0.026 | 0.397 |
| Trustworthiness of computers | 103 | PR‡ | 0.247 | **.006**** |
| Less likely mistakes than humans | 103 | PR‡ | 0.163 | *.05†* |
| Trust computer after mistake | 103 | PR‡ | 0.012 | 0.451 |
| **"I trust a computer to . . ."** | | | | |
| Manage personal finances | 103 | PR‡ | 0.171 | **.042*** |
| Invest in stock market | 103 | PR‡ | 0.219 | **.013*** |
| Manage fitness plan | 103 | PR‡ | 0.135 | *.087†* |
| Plan meals | 103 | PR‡ | 0.127 | *.1†* |
| Diagnose health problems | 103 | PR‡ | 0.185 | **.031*** |
| Drive your vehicle | 103 | PR‡ | 0.208 | **.018*** |

Notes

[1]Question only shown to participants if smartphone usage not equal to never.

[2]Question only shown if navigation app usage not equal to never.

[3]Recoded variable (inverted).

Abbr: PR = Pearson's R, SR = Spearman's Rho.

‡One-sided test. Significance

†p<0.1

*p<0.05

**p<0.01

***p<0.001

## Substantial individual differences in algorithm adherence

As the observed sequences of returns were approximately identical between the human and algorithmic managers, this allowed us to directly compare the participant's investment behavior between the manager types. Hence, we measured each participant's level of algorithm adherence as the difference in their investment rates (i.e. the percentage of rounds in which they decided to invest in the fund) between human and algorithmic managers. The resulting

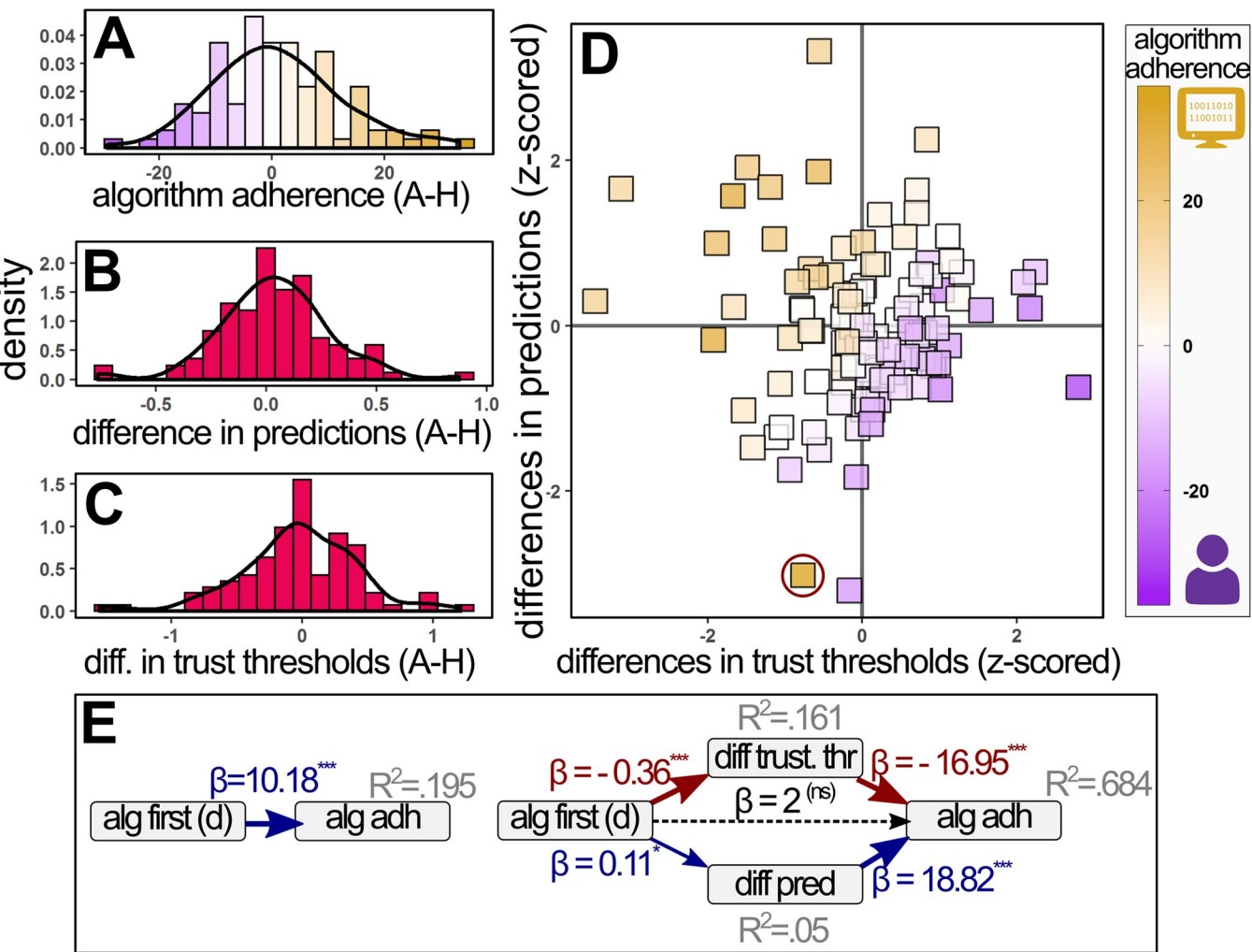

**Fig 3. Experiment results.** Icons indicate the direction of affinity for human or algorithmic fund-managers. **A:** the distribution of observed algorithm adherence values, calculated as the participants' difference in investment rates between algorithm- and human-managed funds. Higher values indicate higher algorithm adherence. **B:** the distribution of participants' differences in predictions, calculated as the participants' difference in mean predicted returns of algorithmic- versus human-managed funds. Higher values indicate a higher relative prediction for the return of funds managed by algorithmic managers. **C:** the distribution of participants' differences in trust thresholds, calculated as the predicted return required for the participant to be indifferent towards investing in an algorithmic manager's fund versus a human manager's fund. Higher values indicate a higher relative threshold against investing in an algorithm-managed fund. **D:** scatterplot depicting the relationship between all three main variables. The x-axis denotes the difference in trust thresholds, the y-axis denotes the difference in predictions and the color indicates the level of algorithm adherence. The red circle indicates an outlier participant. **E:** SEM model results for the direct effect of presentation order on algorithm adherence (left) and mediated model (right).

individual algorithm adherence scores were approximately normally distributed around zero but with a sizable standard deviation (Fig 3A, mean = 0.95, SD = 11.41). We did not observe evidence of an overall difference in mean investment rates between human and algorithmic investment managers (two-sided t[204] = 0.47, p = .643). Roughly equal numbers of participants displayed some degree of algorithm aversion (i.e. a difference value below zero, n = 45) or algorithm appreciation (i.e. a difference value above zero, n = 46). A small minority of participants showed algorithm neutrality (i.e. a difference value equal to zero; n = 12). In line with previous findings, this result suggests that neither algorithm aversion nor appreciation are dominant across the sample, but rather that participants fall on a continuum between the two extremes.

What individual characteristics might underlie these differences in algorithm adherence? Table 1 presents correlations between various demographic measures and algorithm adherence. A detailed version of the questionnaire items and question text is provided in S1 Table. We did not have an a-priori directional hypothesis for the correlation between participants' algorithm adherence and their age, education, gender, and risk aversion. As a result, these correlations were tested using a two-sided test. For all remaining questions we had a directional hypothesis: algorithm adherence was predicted to have positive correlations with participant's smartphone and app usage, reliance on algorithmic video suggestions, positive attitudes towards computers and an increased trust to utilize computers in a range of tasks. As such, all tests were conducted with a one-sided test distribution.

We did not observe a significant relationship between algorithm adherence and age (Pearson $r = -.087$, $t[101] = -0.88$, $p = .38$), education level (Spearman Rho $= .037$, $p = .713$), gender ($F[2,100] = -0.38$, $p = .707$) or risk aversion (Pearson $r = -.151$, $t[101] = -1.532$, $p = .129$). While algorithm adherence did not correlate with items related to general algorithm familiarity (such as smartphone and app usage), we did observe a consistent (although marginal for some items) correlation between algorithm adherence and self-stated trust in computers for a range of tasks (all p's $< .10$ in the predicted direction), as well as two items pertaining to the perceived reliability of computers and their likelihood of producing errors. Furthermore, our measure of algorithm adherence was not correlated with the participants' overall earnings for the experiment (Pearson $r = -027$, $t[101] = -0.27$, $p = .785$), investment rate (Pearson $r = .078$, $t[101] = 0.78$, $p = .44$), forecasting accuracy as measured by participants 'overall mean absolute forecast error (Pearson $r = .114$, $t[101] = 1.16$, $p = .25$) or mean self-reported confidence in their forecasts (Pearson $r = -.015$, $t[101] = -0.15$, $p = .88$). We next explored where either or both of the mechanism discussed above captured a significant proportion of the observed variance in algorithm adherence across the sample.

## Replication of previously reported effects for both predictions and trust thresholds

Previous research has suggested that individual differences in algorithm adherence (partially) result from individual differences in the predicted efficacy of human versus algorithmic partners. Here we measured this difference in predictions by subtracting participants' mean predicted returns for human-managed funds from their mean predicted returns for algorithm-managed funds. The resulting prediction-difference scores were approximately normally distributed around zero (Fig 3B, mean $= 0.04$, SD $= 0.25$). The results of a simple regression model indicated that the participant's difference in predictions significantly predicted their level of algorithm adherence ($\beta = 19.62$, $t = 4.08$, $p < .001$, adj.$R^2 = .179$), whereby participants who held higher predictions for humans as compared to algorithms exhibited greater algorithm aversion, and vice versa. Additionally, the fit of this model substantially increased after excluding one outlier with a residual score larger than 4.5 SD from the mean ($\beta = 25.8$, $t = 6.84$, $p < .001$, adj.$R^2 = .312$).

In line with the perfect automation schema, we hypothesized that this individual variability in predictions would be driven by the differential updating of predictions after trials where the observed return value was *lower* than predicted, but not after trials where the observed value was *higher* than predicted. That is, we expected that situations in which the manager performed worse than expected would give rise to subsequent differences in predictions across manager types. In order test this hypothesis we estimated a triplet of weighted least squares mixed effects models. The dependent variable for each analysis was the participant's round-level updating of predicted return values, defined for each round (t) as the difference between

the predicted returns of the consecutive round (t+1) minus the forecast on round (t). All models included participant-level random intercepts as a random effect. For each model the fixed effects consisted of the round's prediction error (the difference between the round's observed and forecasted return), the participant's measured degree of algorithm adherence, and a dummy variable denoting the manager type (1 = algorithmic manager). The first model included all rounds, while the second and third models included only rounds in which the prediction error was positive or negative, respectively. As a result of the task setup we predicted, and indeed observed, a sizable degree of heteroskedasticity in the model residuals as a function of the round's prediction error: there was substantially more variance in the residuals for rounds in which the absolute value of the prediction error was larger. In order to control for this, we applied a weighted least squares procedure with a weighting function exponentially decaying as a function of the absolute value of the round's prediction error.

Table 2 reports the parameter fits of three models we used to test this difference in updating. In the first model we observed a significant main effect of the trial's prediction error ($\beta = 0.59$, $t = 77.24$, $p < .001$), with no main effects of either algorithm adherence ($\beta = -0.001$, $t = -1.28$, $p > .1$) or whether the manager was an algorithm or a human ($\beta = -0.002$, $t = -0.17$, $p > .1$), and a significant positive interaction between algorithm adherence and the manager type ($\beta = 0.003$, $t = 2.25$, $p < .05$). The second and third model indicated that this interaction was driven by the subset of trials in which the algorithmic manager performed worse than expected

**Table 2. Weighted least squares mixed effects regression models predicting round-by-round updating of predictions.**

**Round-to-round updating of predictions (WLS)**

| | Dependent variable: update prediction between rounds | | |
|---|---|---|---|
| | 1 | 2 | 3 |
| | All Rounds | Above Pred. | Below Pred. |
| Prediction Error | **0.59**\*\*\* | **0.62**\*\*\* | **0.52**\*\*\* |
| | -0.008 | -0.017 | -0.015 |
| Algorithm Adherence | -0.001 | 0.001 | *-0.003*† |
| | -0.001 | -0.002 | -0.002 |
| Algorithmic Manager (D) | -0.002 | 0.011 | -0.008 |
| | -0.013 | -0.018 | -0.017 |
| AA \* Alg. Mngr (D) | **0.003**\* | -0.002 | **0.006**\*\*\* |
| | -0.001 | -0.002 | -0.002 |
| Constant | **0.05**\*\*\* | *0.04*† | -0.02 |
| | -0.012 | -0.023 | -0.021 |
| Participants | 103 | 103 | 103 |
| Observations | 9,476 | 4,198 | 5,252 |
| Log Likelihood | -10,361.70 | -4,376.70 | -5,838.66 |
| AIC | 20,737.40 | 8,767.40 | 11,691.32 |
| BIC | 20,787.50 | 8,811.80 | 11,737.29 |

All models included a participant-level random intercept. Fixed effects consisted of the round's difference between predicted and observed returns (Prediction Error), the participant's measure of algorithm adherence, a dummy variable denoting the type of manager (1 = algorithm) and an interaction between latter two variables. Model 1 included all rounds, model 2 exclusively included round in which the fund's return was above predicted and model 3 exclusively included rounds in which the fund's return was worse than expected. Notes

†p<0.1

\*p<0.05

\*\*p<0.01

\*\*\*p<0.001

(model 3: β = 0.006, t = 3.85, p < .001), but was not observed for rounds in which the algorithmic manager performed better than expected (model 2: β = -0.002, t = -1.11, p > .1). Taken together, these results indicate that participants who were more algorithm averse placed a greater weight on worse-than-expected performance of algorithmic (but not human) managers. Inversely, algorithm appreciative participants placed a lower weight on worse-than-expected performance of algorithmic managers.

Previous research has suggested that individual differences in algorithm adherence (partially) result from individual differences in their relative trust thresholds for human versus algorithmic partners. Here we measured these individual differences in trust threshold via a two-step procedure: first we calculated each participant's trust threshold for investing in human and algorithmic managers separately. These thresholds were defined as the forecasted return value for which the participant was equally likely to invest or not invest in either manager type. These values were estimated by fitting two logistic curves for each participant, predicting the participant's probability to invest on a round given that round's forecasted return value. One curve was fitted for the algorithmically managed funds, whilst the other curve was fitted for the rounds in which the fund was managed by a human. Next, we calculated the level of forecasted return for which this curve had a value of 0.5 (i.e. the level of predicted fund returns for which the participant was indifferent between investing and not investing). Then, we defined each participant's difference in trust thresholds as the difference between these two indifference points, subtracting the human threshold value from the algorithm threshold value. Participants who required a higher prediction before investing in an algorithmically managed fund (relative to a human-managed fund) had a trust threshold difference above zero. In contrast, participants who required a lower prediction before investing in an algorithmically-managed fund had a trust threshold value below zero.

The observed individual differences in trust thresholds were approximately normally distributed, albeit with a slight negative skew (Fig 3C, mean = -0.024, median = -0.021, SD = 0.44). In line with previous literature, the results of a simple regression indicated that these differences in trust thresholds significantly predicted algorithm adherence on the individual level (β = -17.83, t = -9.54, p < .001, adj.$R^2$ = .469), such that algorithm averse participants required a larger predicted return from an algorithm-managed fund (as compared to a human-managed fund) before committing to an investment. Inversely, algorithm appreciative participants required a lower predicted return from an algorithm-managed fund before deciding to invest. Removal of one participant with an excessive residual value (more than 3 SD from the mean) did not meaningfully change these results (β = -17.37, t = -9.73, p < 0.01, adj. $R^2$ = .481).

## Predictions and thrust threshold are uncorrelated and independently effect algorithm adherence

The current experiment was designed to test whether individual differences in predictions and individual differences in trust thresholds exert independent effects on algorithm adherence, or whether they are underlain by the same process. While we observed that differences in algorithm adherence were significantly predicted by both the differences in predictions and in trust thresholds, we did not observe a significant correlation between these two effects (Fig 3D, Pearson r = 0.002, t[101] = 0.03, p = .98). This independence between the two effects was further corroborated by estimating a series of regression models in which individual algorithm adherence was predicted by the z-scored versions of both effects as main effects (Table 3). Results indicated that the sum of the variance explained by the two regressions containing a single mechanism (model 1: adj.$R^2$ = .178, model 2: adj.$R^2$ = .469) was roughly equal to the

**Table 3. OLS regressions comparing the relative strengths of the two observed differences on algorithm adherence.**

| Comparison Standardized Effects (OLS) | | | | | |
|---|---|---|---|---|---|
| | Dependent Variable: Algorithm Adherence | | | | |
| | 1 | 2 | 3 | 4 | 5 |
| A: Difference in | 4.93*** | | 4.95*** | 6.05*** | 7*** |
| predictions (z-scored) | -1.02 | | -0.66 | -0.68 | -0.54 |
| B: Difference in | | -7.86*** | -7.87*** | -8.75*** | -8.19*** |
| trust thresholds (z-scored) | | -0.82 | -0.66 | -0.66 | -0.51 |
| A * B | | | | 2.87*** | 2.06*** |
| | | | | -0.73 | -0.57 |
| Constant | 0.95 | 0.95 | 0.95 | 0.94 | 0.53 |
| | -1.02 | -0.82 | -0.66 | -0.62 | -0.48 |
| Obs | 103 | 103 | 103 | 103 | 102 |
| $R^2$ | 0.187 | 0.474 | 0.662 | 0.708 | 0.814 |
| Adj. $R^2$ | 0.178 | 0.469 | 0.655 | 0.699 | 0.808 |

All five models predicted participants' algorithm adherence as the dependent variable. In order to compare the relative strength of both differences, the participants' difference in predictions and the difference in trust thresholds were z-scored prior to analysis. Models 1 through 4 included all 103 valid participants. Model 5 excluded a single participant with a residual score over 6 SD's above the mean residual score. Notes

†$p<0.1$

*$p<0.05$

**$p<0.01$

***$p<0.001$.

model fit of the regression containing both mechanism (model 3: adj.$R^2$ = .655), indicating that each mechanism captures a different subset of the total variation in algorithm adherence. Model 4 provides the results of a full factorial model indicating a significant main effect of both the participants' difference in predictions (β = 6.05, t = 8.9, p < 0.001) and their difference in trust thresholds (β = 8.75, t = 13.28, p < 0.001), as well as a significant interaction between the two (β = 2.87, t = 3.95, p < 0.001). Overall, the model captured a sizable portion of the individual variance in algorithm adherence (adj.$R^2$ = .699). The relative sizes of the standardized coefficients suggest that the trust threshold effect is a more influential driver of algorithm adherence than is participants' difference in predictions. However, it should be noted that this difference substantially dissipates when excluding a participant with an extreme difference score on trust thresholds (model 5).

## Relative familiarity determines the sign of algorithm adherence

Finally, our experimental design allowed us to explore whether the order in which one encounters human or algorithmic partners can impact decision-making. Interestingly, we observed a significant order effect such that participants who first interacted with an algorithm-managed fund were on average algorithm appreciative (M = 6.88, SD = 10.8), while those who first interacted with a human-managed were on average algorithm averse (M = -3.3, SD = 9.91, t[101] = 4.95, p < .001). The results of a structural equation model indicated that this effect of the initial manager type was fully mediated by the participant's differences in predictions and trust thresholds (Fig 3E, $\chi^2$ [1] = 1.1, p = .295, RMSEA = .031, SPMR = 0.04). This mediation effect was especially pronounced for the difference in trust thresholds, with the order effect explaining 16.1% of the observed variance in this differenced. In other words, participants who were first exposed to an algorithmic manager had a lower relative trust threshold

for algorithms than for humans throughout the experiment. In turn, this decreased trust threshold resulted in a higher degree of algorithm appreciation. Inversely, those who were first exposed to a human-managed fund held a higher trust threshold for algorithmic managers than for human managers, resulting in a higher degree of algorithm aversion.

## Discussion

Our novel experimental paradigm replicates earlier findings on the topic of algorithm adherence and reveals a number of notable findings. Firstly, the experimental data supports the idea of substantial variation in participants' tendencies towards either algorithm aversion or algorithm appreciation. We observe that individuals fall across a substantial range of algorithm adherence, with roughly equal numbers displaying algorithm aversion and algorithm appreciation respectively. Instead of a preponderance of either algorithm aversion or appreciation, these findings support a more nuanced story where most participants' algorithm adherence falls on a continuum between these extremes. In terms of the proposed mechanisms for this variation, we first replicated the previously observed differences in predictions regarding the efficacy of algorithms: participants who had a higher average predicted performance for algorithmic than for human managers were relatively more likely to invest in algorithmically managed funds. Importantly, we observe direct evidence this difference in predictions resulted from a difference in the handling of below-expectation performance of algorithmic managers: participants who held a more positive (negative) prediction of algorithmic performance did so because they placed a lower (higher) weight on the impact of perceived algorithmic errors. Finally, we also replicated the previously observed differences in thresholds towards placing trust in algorithms: participants who had relatively lower trust thresholds for algorithmic than for human managers were more likely to invest in algorithmic managers. These findings ensure general confidence in our results: in line with our hypotheses, both predictions and trust thresholds influenced investment behavior.

The primary result of this experiment is the observation that the two purported processes underlying individual differences in algorithm aversion are distinct and separable processes–that is, differences in predictions constitute a unique and uncorrelated effect from differences in trust thresholds. Both effects independently impacted the level of algorithm adherence, and both mechanisms explain a unique portion of the variance of this adherence. Of the two mechanisms, individual differences in trust thresholds explained a larger proportion of the variance in algorithm adherence (47.7%) than did individual differences in predictions (18.7%). This result is corroborated by the relative size of their standardized regression coefficients (-8.75 versus 6.05), although it should be noted that this difference is substantially reduced when one outlier observation is excluded. The current experiment is the first to document the independence of these mechanisms as causal factors in algorithm adherence. The observation of a significant interaction effect between these two biases suggests that their effect on algorithm aversion are mutually reinforcing (i.e. super-additive): a person with both high differences in predictions and differences in trust thresholds is more algorithm averse than can be explained by either mechanism alone.

In addition to these main results, we observed an (exploratory) order effect for the initial manager type. Participants who were first exposed to a human manager were on average algorithm averse, whilst those who initially viewed an algorithmic manager demonstrated, on average, algorithm appreciation. The results of a structural equation model indicated that this order effect was fully mediated by the two proposed mechanisms, with differences in trust thresholds being the predominant mediating effect. This result suggests that participants had a lower threshold towards placing trust in the initially observed manager type. Note that this

order effect cannot be readily explained by participants becoming more familiar with the task over time, as such a learning effect would predict that initial manager type would be construed as relatively *more* uncertain–and hence result in a higher trust threshold for the initial manager type. What might explain this order effect?

Although not a-priori predicted, these results are in line with the existing literature on the status quo bias [28]. After being exposed to their original manager, participants may perceive managers of this type to be the status quo. Any deviations from such a status quo are then perceived to be more uncertain [29]. Accordingly, participants then require a higher prediction before committing trust to this new manager, resulting in a higher relative trust threshold for this manager type. Although further research is needed to confirm these findings, this mechanism may explain why some types of algorithmic support systems appear to be more readily used than other types: in novel tasks in which the user has no pre-algorithmic exposure (e.g. web searches) users may be more comfortable with algorithmic support systems as opposed to tasks in which the user has been typically accustomed to relying on human support (e.g. self-driving cars). To the best of our knowledge, this is the first time such a status quo bias has been observed for algorithm adherence. If replicated in later experiments, these results may hold interesting policy implications, as they suggest that user's algorithm adherence can at least partially be improved through increased familiarity with algorithmic systems.

Although the current experimental design differs significantly from earlier research, we are confident in the validity of our measurements for three reasons. First, we did not observe a correlation between algorithm aversion and game earnings, forecasting accuracy, or self-reported confidence level, indicating that our measurement of algorithm adherence is not driven by mundane factors such as task comprehension. Second, we did not observe a correlation between algorithm adherence and overall investment rate or with independently assessed risk-taking propensity, suggesting that our measure of algorithm adherence is not confounded by participants' risk preferences. Finally, while results derived from the questionnaire show that our measure of algorithm adherence does not correlate with self-reported smartphone or app usage, it does consistently correlate, in the predicted direction, with self-stated willingness to place trust in computers for various decision contexts. Taken together, these results suggest our measure of algorithm adherence captures a valid real-life construct.

The observed independent effects of prediction and trust thresholds on algorithm adherence fit with a number of previously proposed frameworks. First, both mechanisms map closely to a recent systematic review on factors influencing trust in automation [30]. Our definitions of prediction and trust thresholds map closely to first and last layers in their model, namely dispositional and learned trust. Additionally, the existence of independent mechanisms have a conceptual overlap to biological models of reinforcement learning [31]. At the core of the reinforcement learning framework lie two separate functions: a state-value function in which an agent keeps track of the observed environment and is instrumental in forming beliefs about the state of the world, and a policy function which maps an agent's inferred state of the world to an action to undertake. Conceptually these functions are reminiscent of the differences in prediction and trust thresholds, respectively. This similarity potentially provides an interesting bridge between the topics of algorithm adherence and (biologically inspired) learning. Although more research is needed to explore the similarities between the two areas of research, the conceptual overlap of the two independent biases with reinforcement learning suggests a fruitful area for future exploration in potentially uncovering the neural underpinnings of algorithm adherence.

The current research design provides a number of methodological advantages over the existing literature. First, while previous methods have mostly relied on measuring algorithm adherence at an aggregate level, the current approach allowed for measurement of algorithm

adherence, predictions, and trust thresholds on an individual level. As indicated by the results outlined above, this participant-level approach provides a much more detailed understanding of how differences in algorithm adherence are formed, and how they evolve as a function of feedback. Second, previous studies have typically involved limited interactions with an advisor, employing either a single or a small number of rounds (for an exception see [13]). This has limited the exploration of how algorithm adherence might change over time, especially after expectations have been violated, which turn out to be a key component when multiple interactions are allowed. Third, previous studies on algorithm adherence have typically employed a judge-advisor system [32] in which the participant makes a forecast, is provided with human or algorithmic advice, and is then allowed to update her estimate. While this paradigm has proved useful in the study of aggregate algorithm adherence, it may be confounded by participants' overconfidence in their own ability [33–36], for similar arguments see [15]). In our design participants do not directly invest in the experimental market but instead to rely on either a human or algorithmic manager, removing the participants' own (perceived) skill from the equation. Fourth, previous studies have almost exclusively relied on measuring the uptake of algorithmic advice (for an exception see [13]). Modern algorithmic decision support systems increasingly extend beyond the mere provision of advice and instead completely supplant individual agency (as is the case in self-driving cars and robotic traders). As participants here are forced to rely on a manager's investment skill, our design is more aligned with such real-life systems. Finally, the experimental setup allowed for the collection of all data with the provision of incentives, and with a minimal use of deception.

Despite these methodological advantages, the current design does have some potential limitations. While previous studies utilizing an off-line sample typically observe aggregate algorithm aversion, at least one previous set of online experiments [15] has observed a majority of participants to hold algorithm appreciative preferences. This difference may hint at a selection effect, such that online experimental environments may attract participants with a more positive disposition towards interacting with algorithmic agents. As a result, we cannot rule out that the online sample utilized in the current experiment may hold a more positive attitude towards algorithms than a typical off-line sample. Future research is needed to determine the extent to which the current findings generalize to an off-line population. Additionally, as our current experimental design only elicited algorithm adherence in a single task, it does not allow us to determine whether the observed individual differences in predictions and trust thresholds are temporally stable individual traits. Nonetheless, the opposite pattern of algorithm adherence due to the statistically significant order effect suggests that trust thresholds are at least partially determined by a user's prior experiences. These results fit with recent findings suggesting that algorithm adherence is greatly increased when experimental participants are allowed a trivial amount of input in the algorithm's decision-making process [2]. Presumably these effects lead to a lower perceived uncertainty of algorithmic systems, resulting in higher levels of algorithm adherence. Although future research is needed, this finding suggests that the determinants of algorithm adherence are not exclusively fixed traits, but instead are subject to modification. Therefore, that the design of human-algorithm interactions could potentially be tailored to promote trust and to minimize the effect of perceived errors.

In terms of practical relevance, this independence of both main effects implies that that there is likely no "silver bullet" intervention when attempting to increase users' adherence to algorithmic support systems, as their (dis)like for algorithms may be derived from different underlying sources. Any intervention targeted at reducing either effect may therefore only provide a partial improvement of algorithm adherence at best, or at worst even aggravate the problem by unintentionally enhancing the opposing effect. As a hypothetical example: while increased exposure to an imperfect algorithmic agent may decrease a user's relative trust

threshold through increased familiarity with the agent, this policy will also expose the user to a larger number of observed errors. As a result, the positive effect of familiarity may unintentionally be offset by an increase in negative predictions of future performance. Depending on the relative strength of both mechanisms, the resulting net effect may even lead to an overall increase in algorithm aversion. Further research could usefully determine which policies (or combinations of individually targeted interventions) can yield an optimal approach to improve algorithm adherence, and this study provides an important starting point for investigating this timely societal issue.

## Supporting information

**S1 Table. Extended details of the questionnaire data.**
(DOCX)

## Acknowledgments

The authors are grateful to the helpful comments from the members present in the Idealab of the chair of finance (Institute for Management Research), the D2P2 lab (Behavioral Science Institute) and the Decision Neuroscience Lab (Donders Institute for Brain, Cognition, and Behaviour), all part of the Radboud University Nijmegen.

## Author Contributions

**Conceptualization:** Achiel Fenneman, Joern Sickmann, Alan G. Sanfey.

**Data curation:** Achiel Fenneman.

**Formal analysis:** Achiel Fenneman.

**Funding acquisition:** Joern Sickmann, Thomas Pitz.

**Investigation:** Achiel Fenneman, Joern Sickmann, Thomas Pitz, Alan G. Sanfey.

**Software:** Achiel Fenneman.

**Supervision:** Joern Sickmann, Alan G. Sanfey.

**Writing – original draft:** Achiel Fenneman.

**Writing – review & editing:** Joern Sickmann, Thomas Pitz, Alan G. Sanfey.

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
