## [Decision Letter · Decision Letter 0]

2 Feb 2021

Two distinct and separable processes underlie individual differences in algorithm adherence: differences in predictions and differences in trust thresholds

PONE-D-20-40227

Dear Dr. Fenneman,

We’re pleased to inform you that your manuscript has been judged scientifically suitable for publication and will be formally accepted for publication once it meets all outstanding technical requirements.

Kind regards,

Lidia Adriana Braunstein, PhD

Academic Editor

PLOS ONE

Additional Editor Comments (optional):

Reviewers' comments:

Reviewer's Responses to Questions

**Comments to the Author**

1. Is the manuscript technically sound, and do the data support the conclusions?

Reviewer #1: Yes

2. Has the statistical analysis been performed appropriately and rigorously? 

Reviewer #1: Yes

3. Have the authors made all data underlying the findings in their manuscript fully available?

Reviewer #1: Yes

4. Is the manuscript presented in an intelligible fashion and written in standard English?

Reviewer #1: Yes

5. Review Comments to the Author

Reviewer #1: The text begins by establishing the existence of a variability in the degree of use of algorithmic systems. This variability is determined by two different factors: the different forecasts made by agents regarding the effectiveness of these systems and the confidence limits that agents have in these systems. The purpose of the article is to determine whether these two factors are correlated or not.

The results obtained by the experiment present a great potential for researchers in the area. In general, the article seems to bring important answers to researchers, not only in the field of Computing, but also in Economics, especially for those who work with Game Theory.

6. PLOS authors have the option to publish the peer review history of their article (what does this mean?). If published, this will include your full peer review and any attached files.

Reviewer #1: No

---

## [Editor Report · Acceptance letter]

11 Feb 2021

PONE-D-20-40227 

Two distinct and separable processes underlie individual differences in algorithm adherence: differences in predictions and differences in trust thresholds 

Dear Dr. Fenneman:

I'm pleased to inform you that your manuscript has been deemed suitable for publication in PLOS ONE. Congratulations! Your manuscript is now with our production department. 

Kind regards, 

on behalf of

Dr. Lidia Adriana Braunstein 

Academic Editor

PLOS ONE